# Seasonal Azithromycin Use in Paediatric Protracted Bacterial Bronchitis Does Not Promote Antimicrobial Resistance but Does Modulate the Nasopharyngeal Microbiome

**DOI:** 10.3390/ijms242216053

**Published:** 2023-11-07

**Authors:** Simon J. Hardman, Fiona M. Shackley, Kelechi Ugonna, Thomas C. Darton, Alan S. Rigby, Debby Bogaert, Justyna M. Binkowska, Alison M. Condliffe

**Affiliations:** 1Department of General Paediatrics, Chesterfield Royal Hospital NHS Foundation Trust, Chesterfield S44 5BL, UK; 2Department of Paediatric Immunology, Allergy and Infectious Diseases, Sheffield Children’s Hospital NHS Foundation Trust, Sheffield S10 2TH, UK; fiona.shackley@nhs.net; 3Department of Paediatric Respiratory Medicine, Sheffield Children’s Hospital NHS Foundation Trust, Sheffield S10 2TH, UK; k.ugonna@nhs.net; 4Division of Clinical Medicine, School of Medicine and Population Health, University of Sheffield, Sheffield S10 2RX, UK; t.darton@sheffield.ac.uk (T.C.D.); a.m.condliffe@sheffield.ac.uk (A.M.C.); 5Hull York Medical School, University of Hull, Hull HU6 7RX, UK; asr1960@hotmail.com; 6Department of Paediatric Immunology and Infectious Diseases, Wilhelmina Children’s Hospital and University Medical Center Utrecht, 3584 CX Utrecht, The Netherlands; d.bogaert@ed.ac.uk; 7Centre for Inflammation Research, Institute for Regeneration and Repair, University of Edinburgh, Edinburgh EH8 9YL, UK; justyna.binkowska@ed.ac.uk

**Keywords:** paediatrics, antimicrobial resistance, protracted bacterial bronchitis, azithromycin, microbiome

## Abstract

Protracted bacterial bronchitis (PBB) causes chronic wet cough for which seasonal azithromycin is increasingly used to reduce exacerbations. We investigated the impact of seasonal azithromycin on antimicrobial resistance and the nasopharyngeal microbiome. In an observational cohort study, 50 children with PBB were enrolled over two consecutive winters; 25/50 at study entry were designated on clinical grounds to take azithromycin over the winter months and 25/50 were not. Serial nasopharyngeal swabs were collected during the study period (12–20 months) and cultured bacterial isolates were assessed for antimicrobial susceptibility. 16S rRNA-based sequencing was performed on a subset of samples. Irrespective of azithromycin usage, high levels of azithromycin resistance were found; 73% of bacteria from swabs in the azithromycin group vs. 69% in the comparison group. Resistance was predominantly driven by azithromycin-resistant *S. pneumoniae*, yet these isolates were mostly erythromycin susceptible. Analysis of 16S rRNA-based sequencing revealed a reduction in within-sample diversity in response to azithromycin, but only in samples of children actively taking azithromycin at the time of swab collection. Actively taking azithromycin at the time of swab collection significantly contributed to dissimilarity in bacterial community composition. The discrepancy between laboratory detection of azithromycin and erythromycin resistance in the *S. pneumoniae* isolates requires further investigation. Seasonal azithromycin for PBB did not promote antimicrobial resistance over the study period, but did perturb the microbiome.

## 1. Introduction

Chronic cough (for more than four weeks) is very common in children. A recent Swiss cross-sectional study reported a prevalence of 12% in 6–9 year olds [1]. Protracted bacterial bronchitis (PBB) has been identified as one of the commonest causes of chronic cough in young children referred to paediatric specialists, accounting for up to 40% of cases [2,3]. Other causes include asthma and gastro-oesophageal reflux, depending on the population studied [4]. Strictly defined, PBB is characterised by an isolated persistent wet cough exceeding four weeks duration (commoner in the winter months), growth of a single bacterial species on bronchoalveolar lavage and resolution of the cough in response to two weeks of an appropriate oral antibiotic [5]. Due to the challenges of undertaking paediatric bronchoscopy, the diagnosis is often made based on patient symptoms, rather than by obtaining microbiological samples from the lower airways. Recurrent episodes of PBB have been associated with the development of bronchiectasis in up to 9.6% of cases [6]. Other secondary impacts include significant worry and sleepless nights for parents as well as being a cause for multiple medical appointments [7]. 

Azithromycin, a broad-spectrum azalide, has both antimicrobial and anti-inflammatory properties [8]. Azithromycin prophylaxis has been used in a number of paediatric respiratory diseases. For example, in patients with cystic fibrosis (CF), long-term azithromycin use for 6 to 33 months has been found to reduce the number of pulmonary exacerbations, the number of days hospitalised for pulmonary exacerbations and, in some studies, to improve FEV1 [9,10]. Its use is recommended in the national cystic fibrosis consensus guidelines [11]. There is a growing evidence base for its use in other adult and paediatric respiratory conditions to reduce recurrent pulmonary exacerbations including non-CF bronchiectasis, as recommended by the British Thorax society guidelines [12] and primary ciliary dyskinesia [13]. However, several key trials investigating long-term macrolides in CF, non-CF bronchiectasis, chronic obstructive pulmonary disease and asthma have also reported significant increases in macrolide-resistant respiratory pathogens [14,15,16,17]. In one study, a significant increase in the proportion of macrolide-resistant organisms was still evident at 6 months after completing treatment with azithromycin [18]. 

Antibiotic use is also known to have further adverse impacts on the microbiome [19]. In health, the microbiome confers beneficial effects including modulation of the host’s immune system, providing colonisation resistance and producing metabolites that interact with systemic metabolism of the host [20,21]. The composition of the respiratory microbiome develops throughout childhood and varies markedly between healthy children and those with respiratory conditions. For example, the presence of some genera within the nasopharynx such as *Prevotella* spp. and *Leptotricha* spp. has been strongly associated with subsequent development of upper respiratory tract infections [22]. The persistence of certain bacterial genera within the nasopharyngeal microbiome in children aged 2–13 months has been significantly associated with increased risk of developing asthma [23], as has exposure to multiple courses of antibiotics [24]. Few studies have investigated the microbiota of children with PBB. One early study demonstrated that children with CF, non-CF bronchiectasis and PBB had similar core microbiota, perhaps re-enforcing the possibility of pathophysiological links between these conditions [25]. In the lower airways, children with PBB have been found to have a lower alpha diversity (within-sample diversity) than healthy controls and changes in the community composition with significant predominance of *Haemophilus* and *Neisseria* spp. [26]. 

Despite limited evidence for its use in PBB, seasonal azithromycin is often prescribed continuously over the winter months as a strategy to reduce infection burden and perhaps the risk of developing bronchiectasis. We hypothesise that such azalide exposure would promote macrolide-resistant nasal flora and perturb the microbiome. In this prospective cohort study, we aimed to investigate the impact of seasonal azithromycin use on antimicrobial susceptibility of common upper respiratory tract pathogens, and diversity in the nasopharyngeal microbiome in this high-risk patient population. 

## 2. Results

A total of 50 children were recruited who had experienced at least one episode of clinically diagnosed PBB over the previous 18 months. Of these, 25 were commenced on seasonal azithromycin (10 mg/kg/dose thrice weekly) over the winter months as deemed clinically necessary by their respiratory consultant. Azithromycin was not deemed necessary in the second group of 25 children who were used as a comparison group. This was due to the fact they had not had more than three exacerbations over the preceding 12 months as is outlined in the local guideline for the management of PBB [27]. 

Two winter recruitment periods were required to enrol the proposed number of children. The first recruitment period between October 2018 and March 2019 (20 children split equally between the two groups). These children had nasopharyngeal swabs collected 3 monthly for 12 months. The second recruitment period was over the same months the following year, 2019–2020 (30 children split equally between the two groups). These children were swabbed 4 monthly over a 12–20 month period. Seven children did not complete this study, four from the azithromycin group and three from the comparison group (Figure 1).

Children in the azithromycin group tended to be older compared to those in the comparison group, and tended to be more likely to have had a microbiological diagnosis associated with their PBB. Given the opportunistic nature of the recruitment process and non-randomised allocation of the treatment (clinician designated), we compared the baseline characteristics of participants in each group. These differences were not statistically significant, and the groups were otherwise comparable at baseline (Table 1). 

### 2.1. Antibiotic Exposure

Participants in the azithromycin group at the baseline timepoint were found to have been prescribed more courses of antibiotics prior to randomisation than those in the comparison group, with an incidence rate ratio of 1.38 (95%CI 1.02–1.89), *p*= 0.037 (Chi^2^-test). However, participants in the azithromycin group were also somewhat older, and when corrected for age, the difference was not significant, *p* = 0.12. Of the participants in the comparison group, 14 out of 25 (56%) had previously been prescribed at least one course of macrolide antibiotics in their lifetime, as per their GP and hospital inpatient prescription records (Table 2). In the azithromycin group, 13 out of 25 (52%) had previously been prescribed at least one course of macrolide antibiotic or at least one seasonal course of macrolide prophylaxis. 

During this study, participants in the azithromycin group were prescribed, thrice weekly, oral azithromycin at 10 mg/kg/dose for a median duration of 7 months (range: 3 days to 17 months; (IQR 4.5–12 months)). 

The azithromycin compliance diaries were completed by 8 out of 25 families (32% response rate). A total of 15 doses were recorded as not given over a combined period of 40 months (overall, 3% of doses missed). Of these, 8/15 doses were missed during school holidays. With additional ethical approval, we reviewed prescription records to improve the data quality relating to antibiotic exposure. 

Analysing data collected from families and GPs and collated with hospital prescription records, there was no statistical difference between the number of acute courses of antibiotics prescribed for presumed respiratory infections during this study between the azithromycin group and the comparison group, 33 versus 54, respectively, IRR 0.69 (95%CI 0.41–1.14), *p* = 0.15. Of the acute courses of antibiotics prescribed, 6% were macrolides in the azithromycin group compared to 9% prescribed macrolides in the comparison group. Co-amoxiclav was the most frequently prescribed antibiotic in both groups (45/87, 51% of prescriptions) with amoxicillin being the second most common antibiotic used (35/87, 40%), (Appendix A). There were fewer prescriptions for courses of antibiotics >2 weeks in duration to treat presumed PBB exacerbations in the azithromycin group compared to the comparison group, 11 vs. 19, respectively; again, this did not achieve statistical significance, IRR 0.65 (0.27–1.55), *p* = 0.33. 

### 2.2. Nasopharyngeal Bacterial Culture Results

A total of 50 baseline nasopharyngeal swabs were collected and 141 follow up swabs were collected during the study period. Of the baseline swabs, 19 of 25 (76%) in the comparison group had a positive growth of at least one bacterial isolate and 16 of 25 (64%) in the azithromycin group (Figure 2). *S. pneumoniae* and *H. influenzae* were the two most cultured isolates 56/151 and 54/151, respectively. Of the *S. pneumoniae* isolates, 54/56 were resistant to azithromycin and 19/54 of the *H. influenzae* isolates. 

During the study period, 49 of the 77 (64%) nasopharyngeal swabs collected in the comparison group yielded a positive growth, with 26 of 64 (41%) from those in the azithromycin group (see Appendix A for further details), which was significantly lower compared to the comparison group, IRR 0.58 (95%CI 0.34–0.98) and *p* = 0.044. 

### 2.3. Phenotypic Antibiotic Resistance Testing

A total of 166 bacterial isolates were tested for azithromycin susceptibility by measuring MICs (minimum inhibitory concentration), from which susceptibility or resistance was interpreted using established cutoffs (E-Test, Biomerieux, Chineham Gate, UK) [28]. At baseline, 14 of the 19 (74%) baseline swabs with bacterial growth in the comparison group had azithromycin-resistant bacteria compared to 11 of the 16 swabs (69%) in the azithromycin group. During the study period after the baseline swabs had been collected, 34 of the 49 (69%) swabs with bacterial growth in the comparison group had azithromycin-resistant bacteria present with 19 of the 26 (73%) swabs in the azithromycin group (Figure 3). Of the bacterial isolates, there was a non-significant trend towards fewer azithromycin-resistant bacteria in the azithromycin group with an IRR of 0.63 (95%CI 0.37–1.06) *p* = 0.083. Adjusting for the number of bacterial isolates per swab, this lack of significance remained (*p* = 0.68). Adjustment for the actual time participants took part in this study, ranging from 12 to 20 months due to COVID-19 restrictions, (*p* = 0.67), did not change these results.

Azithromycin resistance was predominantly driven by resistant *S. pneumoniae* with only two isolates being classed as susceptible to azithromycin (Figure 4). Within the comparator group, 21 of 25 participants had azithromycin-resistant *S. pneumoniae* present and 13 of 25 in the azithromycin group. However, in most cases, the resistance could be regarded ‘low level’ with MICs above 0.5 mg/l but below 8 mg/L. Only six highly resistant isolates were identified, with MICs > 256 mg/L. These were identified in two participants in the comparison group and three within the azithromycin group. When reviewing all the swabs collected, the majority of *H. influenzae* isolates were classed as azithromycin susceptible, with some ‘low-level’ resistance (Figure 4). Azithromycin-resistant *H. influenzae* were found in 10 of 25 participants in the comparison group and 2 of 25 in the azithromycin group. See Appendix A for the MIC distributions of *M. catarrhalis* and *S. aureus*.

Extended phenotypic susceptibility testing was carried out on 150 isolates available for further analysis. Table 3 reports the azithromycin bacterial isolates tested using disc diffusion. A total of 54 *S. pneumoniae* isolates were found to be azithromycin resistant using an agar-gradient diffusion method, E-test strips. Of these, 46 were available for extended phenotypic susceptibility testing with only five found to be erythromycin resistant using the same agar-gradient diffusion method, E-test strips, see Figure 5. These five isolates were all highly resistant to azithromycin (MIC > 256 mg/L using E-test strips). Three of these five were beta lactam resistant using oxacillin disc diffusion screening (all three, 6 mm). Of the 54 *H. influenzae* isolates, only 19 were azithromycin resistant.

The two tables report the antibiotic susceptibility of Table 3a *S. pneumoniae* and Table 3b *H. influenzae* to a panel of clinically relevant antibiotics. There was no resistance of *S. pneumoniae* to chloramphenicol or levofloxacin (30 isolates were, however, not tested for levofloxacin susceptibility due to a protocol error). There was no resistance of *H. influenzae* to either tetracycline or ciprofloxacin. 

### 2.4. Genotypic Resistance of S. pneumoniae Isolates

The phenotypic resistance profiles demonstrated a major and unexpected discrepancy between erythromycin and azithromycin resistance amongst the *S. pneumoniae* isolates, 3% vs. 98% resistance, respectively. In order to investigate this further, DNA extracted from *S. pneumoniae* isolates underwent genomic sequencing. Forty *S. pneumoniae* isolates of the initial 58 were sequenced to provide genomic data (see Appendix A for quality assessment). Of these, 37 were used in the final analysis. Of the three samples not included, two samples had total contig lengths double that of *S. pneumoniae* suggesting possible contamination. One sample had been incorrectly labelled and was identified as *H. influenzae*. 

Mass screening of 1236 contigs (37 isolates) for antimicrobial resistance genes was carried out, including 4 available isolates with high-level resistance to azithromycin. Erm(B) and tet(M) were present in three of the four high-level azithromycin-resistant isolates (MIC > 256 mg/L). Msr(D) and mef(A) were present in the fourth isolate. Only one isolate was a pneumococcal vaccine serotype (19A) and was not associated with any identifiable resistance genes. 

Pangenome assessment allowed comparison of the non-core genomes. When selecting samples with high-level azithromycin resistance (MIC ≥ 256 mg/L) compared to all other samples, one hypothetical protein sequence was positively identified in the 4 highly resistant samples and not identified in any of the other 33 samples; sensitivity 100%, specificity 100%. The Bonferroni *p* value was used as a conservative approach to control for multiple comparison corrections; *p* < 0.00001. The sequence of this hypothetical protein was identified as (letters represent amino acids):

‘MTKELQSSRYIVISFLVREMGIDIVEAISLMAELEKSGLVRLESSGDLILKELGGAL’

The Streptococcus Pneumoniae Comparative System and PaperBLAST databases were searched and identified the protein as SPN23F13170 polypeptide, a putative uncharacterised protein consisting of 57 amino acids. This protein has been previously identified in *S. pneumoniae* on a gene acquired horizontally on the antibiotic resistance conferring Integrative and Conjugate Element (ICE) ICESp23FST81. Although its role is uncertain, it flanks the element in its linear form and comes into close association when in plasmid form [29].

No other sequences were found that were specific to a group of samples when looking at the clinical indices of sex, azithromycin exposure or not, nor when groups of different azithromycin MICs was examined.

### 2.5. Nasopharyngeal Microbiome

A total of 100 nasopharyngeal swabs were collected from participants recruited in the second year of recruitment and used for microbiome analysis. Of these, 54/100 were collected from participants in the comparison group and 46/100 from those in the azithromycin group. In this cohort, those in the azithromycin group were treated with seasonal azithromycin for a median duration of 12 months (IQR 7.25–12 months), which is longer than the expected 6-month exposure over the winter months. This was due to the fact that many children remained on azithromycin beyond the planned winter duration during the beginning of the COVID-19 pandemic. 

In preprocessing, seven samples were removed as they did not meet the quality control standards (Appendix A). A total of 2694 ultra rare and 113 contaminant taxa were removed leaving 299 taxa to be used in the final analysis. 

At the baseline timepoint, within-sample (alpha) diversity, as measured by the Shannon index, was no different. At 4 months, when all participants were actively taking azithromycin at the time of swab collection and had been exposed to azithromycin for 4 months, within-sample diversity was significantly lower in samples collected from the azithromycin treated group, Shannon index 0.62 vs. 1.71, *p* = 0.02, T test; Figure 6, Table 4. For the third set of swabs (8 months), by which time it had been anticipated that seasonal azithromycin use would have stopped, two participants in the azithromycin group were still actively taking azithromycin. At the final swab collection, four participants were still actively taking azithromycin. At 8 months and the final swab, there was no longer a significant difference in alpha diversity between the two groups. 

At the final swab collection, four participants were still taking azithromycin and had been for a mean of 9.5 months. Those no longer taking azithromycin at the final swab had been off azithromycin for a mean of 5 months. 

Beta-diversity (between-sample diversity) was not significantly different between groups before the start of seasonal azithromycin nor at the second and third collection timepoints. We only observed a significant difference between groups at the final swab collection. Only at this timepoint did being in the azithromycin group (versus the comparison group) significantly contribute to microbiota dissimilarity and accounted for 14% of the variance in microbiota composition (PERMANOVA; R2 = 0.14, *p* = 0.0049) (Appendix A). On further investigation, active treatment with azithromycin (at the time of swab collection) had a statistically significant effect on microbiota composition when compared to all individuals (both in the azithromycin group and comparison group) not currently taking azithromycin (PERMANOVA, R2 = 0.026, *p* = 0.01), Figure 7.

Pairwise models were used to further delineate this difference. These demonstrated a more striking effect on the microbiota composition when those actively taking azithromycin at the time of swab collection only were compared to those individuals in the comparison group, R = 0.079, *p* = 0. 001, (Appendix A). 

We also analysed our data on an individual taxa level (Appendix A), using differential abundance testing, “metagenomeseq”, to compare the two groups (Appendix A). At baseline, there was a significantly higher abundance of Haemophilus in the azithromycin group (adjusted *p* < 0.001). At the final swab collection, there was a significantly lower abundance of *Moraxella* and *Dolosigranulum pigrum* and greater abundance of *Corynebacterium* compared to the comparison group (adjusted *p* < 0.005).

## 3. Discussion

The efficacy of seasonal azithromycin to reduce exacerbations of PBB and future risk of developing bronchiectasis is currently unknown, as is the impact on antimicrobial resistance and dysbiosis. Hence, the risks and benefits of this treatment approach are difficult to evaluate. A high prevalence of azithromycin resistance was found in both study groups. Despite an unexpected dichotomy between azithromycin and erythromycin resistance in *S. pneumoniae*, we did not find that exposure to seasonal azithromycin promoted azithromycin resistance in the two most prevalent bacterial species, *S. pneumoniae* or *H. influenzae*. Seasonal azithromycin use was associated with a trend to fewer prescriptions of acute courses of antibiotics, this was in a group with likely more severe disease than their compactors who were judged not to require seasonal antibiotics. Seasonal azithromycin did lead to dysbiosis of the nasopharyngeal microbiome. This was transient for within-sample (alpha) diversity whilst participants were actively taking azithromycin. For between-sample (beta) diversity, actively taking azithromycin contributed to microbiota dissimilarity as well as being in the azithromycin group at the study end.

Other studies report increased azithromycin resistance associated with azithromycin use [15,17,30,31]. One of the factors underpinning our results is that baseline azithromycin resistance in both groups and ongoing azithromycin resistance in the comparison group were much higher than expected, even in those who had apparently never previously been exposed to macrolides. This higher than expected azithromycin resistance may be particular to children with PBB, it may represent high community carriage of azithromycin-resistant bacteria, or could reflect a technical issue with the assay measuring azithromycin MIC. UK data from 2020 report just 6.0% of invasive *S. pneumoniae* report being macrolide resistant [32]. This is in keeping with the level of erythromycin resistance in our cohort and importantly we did not find any evidence of increased erythromycin resistance in samples from children receiving seasonal azithromycin prophylaxis.

Unexpectedly, we found a major discrepancy between laboratory detection of azithromycin and erythromycin resistance in the *S. pneumoniae* isolates. By re-culturing a subset of the original azithromycin *S. pneumoniae* and carrying out azithromycin and erythromycin susceptibility testing in parallel, we were able to validate the original results. The agar-gradient diffusion method using E-test strips is a validated and widely used method for determining macrolide resistance. In *S. pneumoniae* it has been reported that there is 98% agreement between E-test strips and a standard microbroth dilution method in determining erythromycin resistance [33]. The use of ‘AB biodisk’ E-test strips were found to falsely overestimate *S. pneumoniae* MICs in the presence of carbon dioxide which lowered the pH of the agar plates [34]. However, we used E-test strips from a different manufacture and adhered to the recommended 4–6% CO2 incubation conditions, hence it is unlikely such technical issues explain this discrepancy, which remains under active investigation. Erythromycin resistance is accepted as a valid surrogate marker for other macrolide resistance [28] and there are few reports comparing resistance of different macrolides in *S. pneumoniae*. Mosleh et al. investigated local *S. pneumoniae* macrolide resistance in Hamadan, Iran. They report that of the 55 *S. pneumoniae* isolates obtained from clinical samples, 25.5% were resistant to erythromycin, 18.2% to clarithromycin and 16.4% resistant to azithromycin (susceptibility testing using E-tests) [35]. Thus whilst some discrepancy is reported, greater erythromycin than azithromycin resistance was reported, perhaps reflecting different patterns of antibiotic usage.

The majority of phenotypic azithromycin resistance found in the *S. pneumoniae* isolates was low level. A two-fold increase in MIC from 0.5 to 1 mg/L may not be clinically relevant given that there is intracellular accumulation of azithromycin in cells such alveolar macrophages and the fact that MICs are set conservatively. The mechanism of the low-level azithromycin resistance in the erythromycin sensitive strains of *S. pneumoniae* remains currently unexplained. No known genetic mutations were identified that underpin this finding, although a hypothetical protein was identified in the four highly azithromycin-resistant isolates (also erythromycin-resistant) and not seen in any of the other isolates tested; the role and function of this protein are not yet clear. It is possible that alterations in gene expression underpin the observed low-level resistance, transcriptional analysis was not undertaken and hence increased expression of resistance factors is a potential explanation. A study of Legionella pneumoniae isolates from China reports 149 strains fully sensitive to erythromycin but 25 exhibited azithromycin resistance [36]. Increased expression of an efflux pump component lpeAB was found to be responsible for the reduced azithromycin susceptibility in all 25 strains. 

Seasonal azithromycin use had an impact on the nasopharyngeal microbiome, both with regard to diversity and community composition. We found a significant reduction in alpha diversity after 3–4 months of azithromycin treatment. This within-sample loss of diversity was transient and had increased again by the end of the study period. Despite a paucity of literature with regard to changes in the respiratory microbiome following long term antibiotic exposure, children with cystic fibrosis have been observed to have reduced bacterial diversity in both upper and lower airways associated with prophylactic co-amoxiclav use [37]. At baseline, *Haemophilus* spp. were overabundant in the ‘to be treated’ azithromycin group and were significantly lower at the time of the first (3–4 month) swab (when almost all patients were still being treated in the azithromycin group). This was a likely beneficial effect in PBB where *H. influenzae* is the most frequently detected pathogen. An accompanying minor increase in *Acinetobacter* spp. (not shown) is likely undesirable but was likewise transient. We also found a significant difference in beta diversity of the nasopharyngeal microbiota whilst actively taking azithromycin compared to those in an unexposed comparison group. Just being in the azithromycin group and having had exposure to azithromycin, significantly contributed to beta diversity differences at the termination of this study (12–20 months). This may represent that differences in beta diversity are slower to develop but persist or be due to the fact that fewer swabs were collected during the 2nd and 3rd round of swabs due to national lockdowns. Associated with this more persistent change in beta diversity, the commensal *Dolosigranulum*, (related to healthy microbiota), continued at lower abundance in the treated group whereas several health-associated *Corynebacteria* spp. were increased, suggesting a mixed impact of desirable and potentially detrimental effects. Whilst most children had stopped taking seasonal azithromycin by the time of final swab collection, the duration between cessation of azithromycin and the final swab did, however, vary considerably due to the COVID-19 pandemic. Hence, our study power was somewhat limited, and further trials would be required to validate these findings.

Our study had a number of limitations. The single-centre design and small sample size may mean bacterial resistance profiles, antibiotic prescribing and changes in the microbiome may not be representative of wider cohorts of children with PBB. The participants prescribed seasonal azithromycin were slightly older and, in our opinion, likely to have had more problematic disease than those in the comparison group given the fact seasonal antibiotics were deemed necessary. Prior engagement with patients/families and clinicians suggested that a randomised allocation would not be acceptable to either group without a pilot observational study as undertaken. The observational design of this study had inherent issues with the ability to control the duration of azithromycin treatment. This was further confounded by the COVID-19 pandemic during which many children initially remained on azithromycin in case it offered any additional benefit. Due to lock down restrictions, the swab timetable was significantly disrupted and many of the planned swabs could not be collected. This impacted the sample size and thus the ability to conduct some of the intended analyses. Given that one of the main concerns in PBB is that of the development of bronchiectasis, permanent damage to the lower airways, it would be of value to understand the changes in the microbiome related to azithromycin in samples from the lung (which would usually require bronchoscopy and lavage in this age group). Due to ethical constraints, we only sampled from the nasopharynx; however, this did allow for longitudinal sampling, which would not have been feasible from the lower airways. It would have been valuable to have a healthy control group as well as the comparison group to look at background nasopharyngeal microbiome and resistance in children not exposed to antibiotics. The impact on gut antimicrobial resistance and microbiome was also not possible.

Despite these limitations, our study informs a number of important questions regarding assessment of risk/benefit for the use of seasonal azithromycin in PBB, an intervention which is not currently supported by randomised controlled trial evidence. We found that seasonal azithromycin use was not associated with an increase in azithromycin-resistant bacterial isolates in the nasopharynx; however, our finding of much higher than expected rates of background azithromycin (but not erythromycin) resistance is of concern in children with PBB (and potentially more widely) and warrants further study. This is particularly pertinent given that erythromycin is often used as a surrogate marker for macrolide resistance. The clinical impact of the perturbations in the nasopharyngeal microbiome we describe, are as yet unknown, although it is encouraging to note that many of the changes were reversible on antibiotic cessation. This has implications for the use of seasonal azithromycin more widely, as infection in many patient groups in whom azithromycin is recommended (e.g., COPD and adult non-CF bronchiectasis) are often concentrated in the winter period. The impact on the microbiome and resistome in other niches, such as the gut, would need to be explored before being able to fully quantify the potential risks of antimicrobial resistance secondary to seasonal azithromycin use. 

## 4. Materials and Methods

### 4.1. Study Design and Recruitment

We conducted a prospective, single-centre (Sheffield Children’s NHS Foundation Trust) observational study with favourable ethical approval and approval from the Health Research Authority (reference numbers 18/WS/0176 and 19/YH/0207). This study was carried out in accordance with the principles of the Declaration of Helsinki and the International Council for Harmonisation Good Clinical Practice guidelines.

A total of 50 children were recruited who had experienced at least one episode over the previous 18 months of clinically diagnosed PBB (criteria as defined by the European Respiratory Society on PBB: chronic wet cough for more than four weeks duration, absence of signs of other causes and cough resolution with two weeks of appropriate oral antibiotics [5]). Exclusion and exclusion criteria are documented in Table 5. Our choice of sample size was pragmatic given this understudied area and a power calculation was not performed. Twenty-five children were recruited as they were about to start seasonal azithromycin (10 mg/kg/dose thrice weekly) over the winter months. This was deemed clinically necessary by their respiratory clinician to prevent further exacerbations of PBB, as is the standard of care set out in the Sheffield Children’s Hospital PBB guideline [27]. The second group of 25 (comparison group) fulfilled the same diagnostic criteria but were judged to not require seasonal azithromycin at enrolment as they had not had more than three exacerbations over the preceding 12 months. 

### 4.2. Sample and Data Collection

Baseline deep nasopharyngeal swabs were taken using a cotton tipped wire swab (Sigma Transwab in liquid Amies medium) at a depth of at least half the distance between the nostril and the ear lobe. These were taken before the first dose of azithromycin in the azithromycin group. In the first cohort of children recruited (October 2018–March 2019), serial swabs were collected every 3 months over a 12-month study period. The second cohort of children recruited during the second winter (2019–2020) had swabs collected every 4 months with a final swab at 12–20 months. The swab timetable and study duration varied somewhat during this study due to patient contact restrictions throughout the COVID-19 pandemic and fewer swabs at an initially planned 6-month interval were obtained.

Participants prescribed seasonal azithromycin were asked to complete a self-reported prescription diary as a measure of compliance. Diaries were reviewed when swabs were collected and families prompted to use them. All participants were given a study booklet to record any courses of antibiotics prescribed over the study period. GP and hospital prescription records were reviewed at the study end to enable a more comprehensive analysis of antibiotic exposure. 

### 4.3. Bacterial Culture and Phenotypic/Genomic Susceptibility Testing

All swabs were cultured in the hospital NHS clinical microbiology laboratory and any growth of clinically important potential respiratory pathogens, *S. pneumoniae*, *H. influenzae*, *S. aureus* and *M. catarrhalis*, was recorded and confirmed by standard techniques [38]. Minimum Inhibitory Concentrations (MICs) were determined for azithromycin susceptibility in identified isolates using an agar-gradient diffusion method (E-test, AB BioMérieux, France). Disc diffusion susceptibility (discs supplied by MAST group Ltd., Bootle, UK) was carried out to a standard set of clinically relevant antibiotics. For *S. aureus* an agar incorporation method was used (Mast Adatabs) using in house agar plates (Oxoid Muller-Hinton powder), (Appendix A). In a subset of samples, an agar-gradient diffusion method was performed using erythromycin E-test strips (LioFilchen). MICs were measured from which erythromycin susceptibility could be interpreted according to breakpoints set by the European Committee on Antimicrobial Susceptibility Testing, 2022 [28]. The exception was for *H. influenzae*. As recommended by EUCAST, an epidemiological cut off was used given the spontaneous cure rates of *H. influenzae* respiratory infections, see Appendix A for MICs used [28].

All *S. pneumoniae* isolates were further sub-cultured. DNA extraction was undertaken via standard spin-column methodology (Qiagen DNeasy blood and tissue kit). A sample of 40 isolates of the initial 58 were sequenced, prioritising those isolates found to be more resistant on phenotypic testing and those from participants with multiple isolates. Whole-genomic sequencing was performed (by Microbes NG) on an Illumina sequencing platform using 2 × 250 BP paired end reads. Assembly and annotation were carried out using the software ‘Kraken’ v1, ‘Burrows–Wheeler Aligner’ 0.7.17 and ‘SPAdes’ 3.15.2 [39,40,41]. Screening contigs for resistance genes was performed in Linux (Ubuntu) using the packages ‘abricate’ (NCBI and ResFinder databases), ‘pneumoCat’, ‘roary’ and ‘scoary’ [42,43,44,45].

### 4.4. The 16s Analysis

In view of the initial findings on bacterial resistance profile, further ethical approval allowed for additional 16s analysis was undertaken on swabs collected during the second round of recruitment. Swabs were transported to the laboratory, cultured and the swab medium frozen at −80 °C within 24 h. The 16s rRNA gene sequencing to characterise the bacterial community composition was performed at the Centre for Inflammation Research within the University of Edinburgh, as previously described [46]. In brief, DNA from the frozen swab medium was extracted using a phenol/bead beater protocol using 0.1 mm zirconium beads. A PCR amplicon library was generated by amplification of the V4 hypervariable region of the 16S rRNA gene with barcoded universal primer pair 533F/806R. Amplicons were quantified by PicoGreen (Thermofisher) and sequenced on an Illumina sequencing platform using 2 × 250 bp paired end reads. The threshold of DNA levels was ≥0.2 pg/µL over negative controls. 

Sequences were processed using a DADA2 (v1.26) bioinformatics pipeline [47]. In summary raw sequences were firstly filtered and trimmed (length threshold 200 base pairs for forward reads, 150 base pairs for reverse reads, truncQ = 2, maxEE = 2). The DADA2 parametric error model and then core sample inference algorithms were run and chimeras removed. Taxonomic annotation was executed using the RDP naïve Bayesian Classifier on SILVA v138 training set [48]. Quality control was conducted using the packages ‘decontam’ (combined function), ‘phyloseq’ in R [49]. A total of 18 negative and positive control samples were included (Appendix A). The list of contaminant taxa were compared to reported potential contaminating genera and removed [50]. Ultra-rare taxa were then removed. Only Amplicon Sequence Variants (ASVs) with a relative abundance of at least 0.1% in at least 2 samples were kept as suggested by Subramanian [51].

### 4.5. Statistical Analysis

This study was reported according to STROBE guidelines [52]. An intention to treat analysis was used. Fisher’s exact test was used for comparing categorical data and a paired T-test for continuous data. Poisson regression analysis was used to report the incidence rate ratio when comparing incident rates between two groups. Due to the presence of higher variation than expected when using Poisson regression, negative binomial regression was used that corrects for over dispersion. For the analysis of 16s data, Alpha diversity (within-sample diversity) was measured using the Shannon diversity index. Differential abundance analysis to compare the two groups was performed using “metagenomeSeq” [53]. Beta diversity (between-sample diversity) was investigated using the Bray–Curtis dissimilarity and visualised using non-metric multidimensional scaling (NMDS) plots. Permutational multivariate analysis of variance (PERMANOVA) was carried out, a distance-based method to test the association of microbial beta diversity with the covariates of interest. PERMANOVA testing was performed using Adonis (‘vegan’ in R, with 10,000 permutations). The significance threshold was set at *p* < 0.05. 

## Figures and Tables

**Figure 1 ijms-24-16053-f001:**
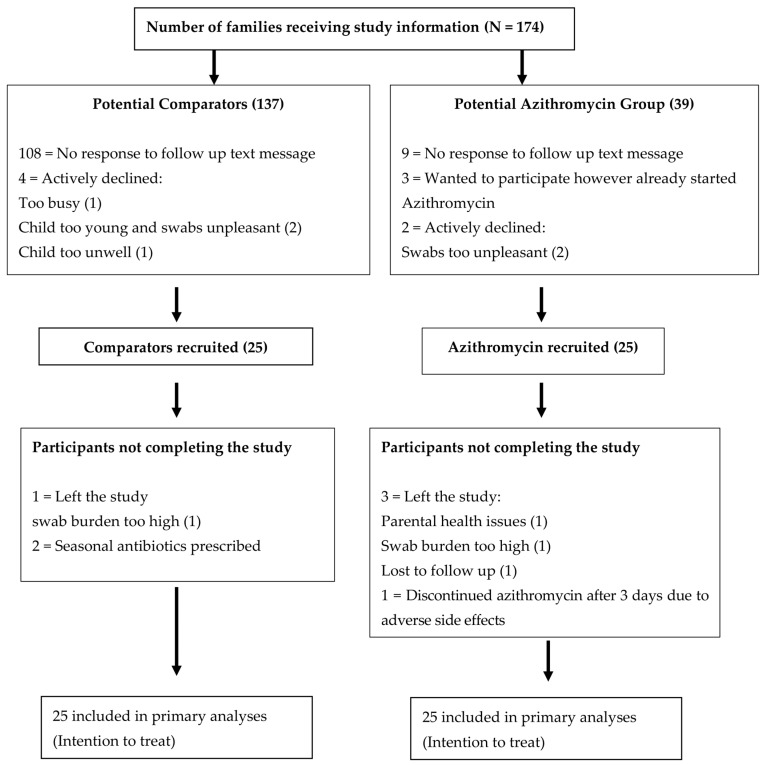
Consort diagram of the recruitment process. All patients seen in the study period who fulfilled the inclusion criteria were given the study information.

**Figure 2 ijms-24-16053-f002:**
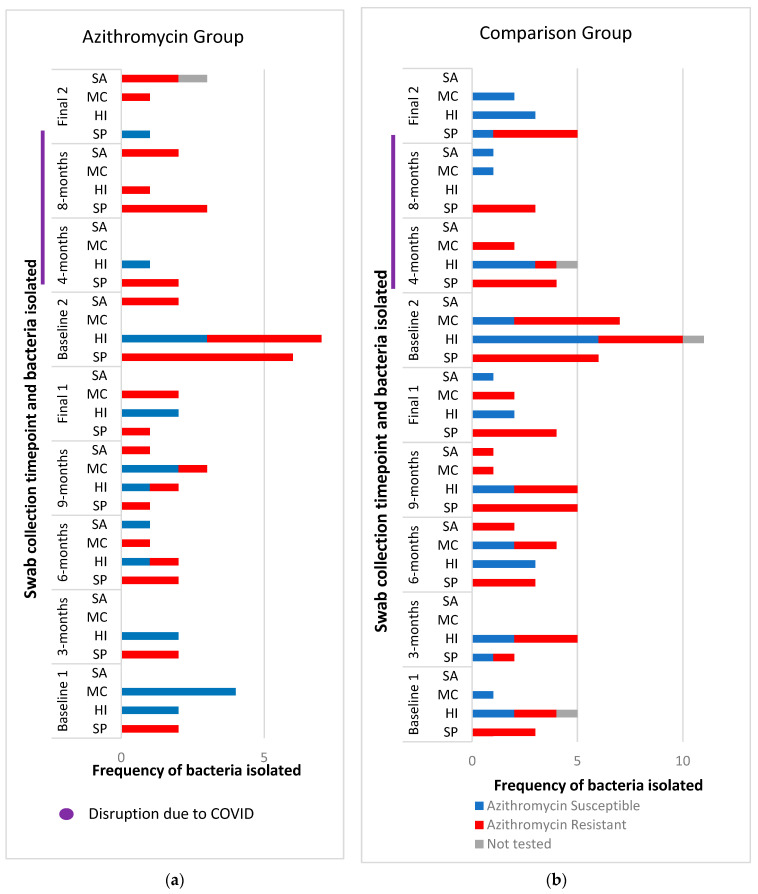
Bacterial species cultured from nasopharyngeal swabs throughout this study. (**a**) Nasopharyngeal bacterial isolates cultured from participants in the azithromycin group and (**b**) comparison group. All swabs were cultured for four common potential respiratory pathogens shown on the Y-axis and tested for azithromycin susceptibility using an agar-gradient diffusion method, E-test strips (EUCAST MIC breakpoints) [28]; SP: *S. pneumoniae* (>0.25 mg/L), HI: *H. influenzae* (>4 mg/L, epidemiological breakpoint), MC: *M. catarrhalis* (>0.25 mg/L), and SA: *S. aureus* (>2 mg/L). The first cohort completed this study at 12 months, Final 1. Baseline 2 swabs were collected at this time (2nd winter of recruitment). Following the Baseline 2 swabs, 4 monthly swabs were collected. The swabs at Final 2 were collected 12–20 months after Baseline 2. The difference in swab intervals was due to significant disruption during the COVID-19 pandemic and national isolation restrictions.

**Figure 3 ijms-24-16053-f003:**
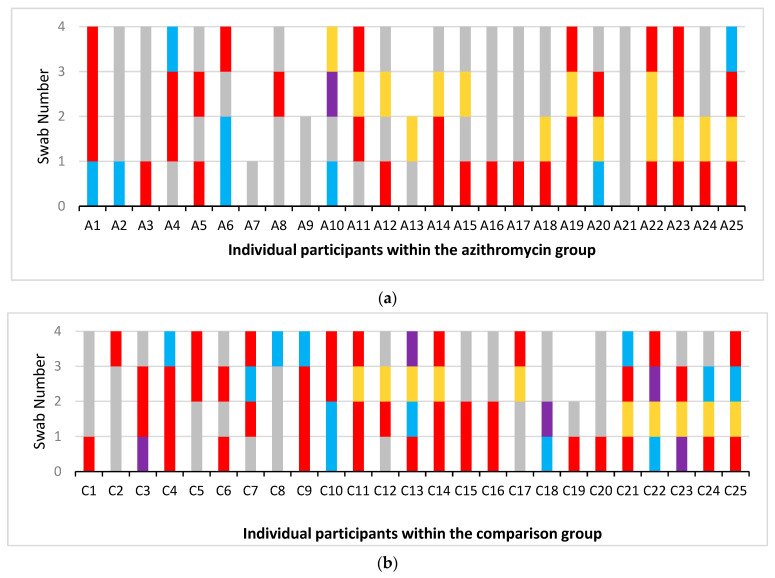
Representation of azithromycin susceptibility for deep nasopharyngeal swab bacterial isolates in the azithromycin group (**a**) and comparison group (**b**). Two cohorts of 25 children with PBB, receiving seasonal azithromycin or not, had sequential swabs taken over the study period. Here, we present the baseline (swabs 1), 3–4-month (swab 2), 8-month (swab 3) and the final swabs. Due the COVID-19 pandemic, few of the planned 6-month swabs were obtained and this timepoint was hence omitted from this representation. Each column on the X-axis represents an individual child. The Y-axis shows each of the swab results with different colours denoting whether an azithromycin-resistant bacterial isolate was present. MICs used to determine azithromycin resistance: *S pneumoniae* > 0.25 mg/L, *H. influenzae* > 4 mg/L, *M. catarrhalis* > 0.25 mg/L *S. aureus* > 2 mg/L [28].

**Figure 4 ijms-24-16053-f004:**
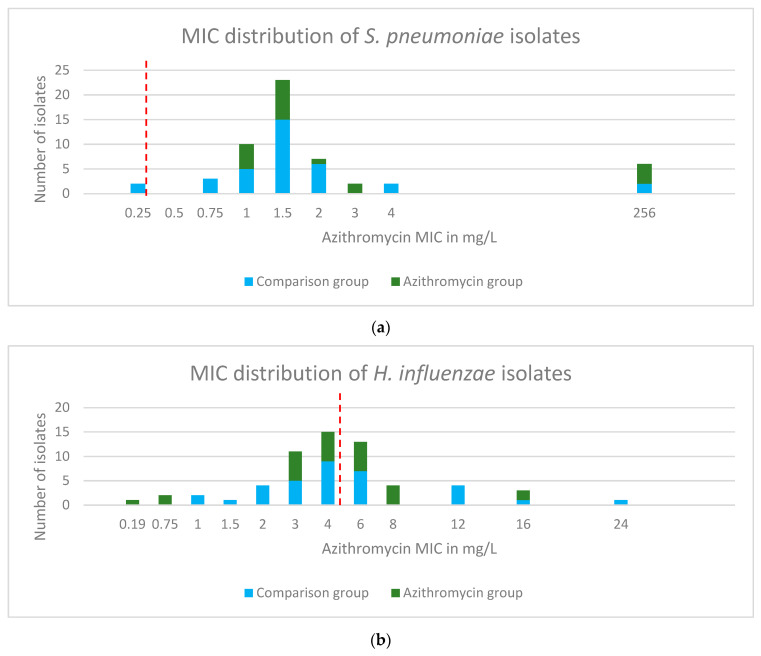
Azithromycin MIC results for bacterial isolates. Azithromycin susceptibility was tested using an agar-gradient diffusion method, E-test strips. Susceptibility was determined using the EUCAST break points for each isolate. (**a**) *S. pneumoniae*; (**b**) *H. influenzae*. Dashed red lines indicate EUCAST breakpoints.

**Figure 5 ijms-24-16053-f005:**
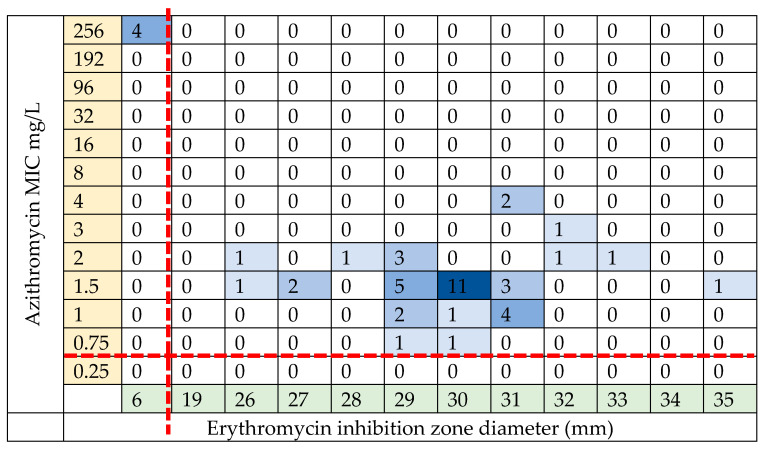
Relationship between azithromycin MIC and erythromycin inhibition zone size in *S. pneumoniae* isolates. The plot shows the relationship between the azithromycin MIC on the X-axis and erythromycin disc diffusion inhibition zone on the Y-axis. The squares are coloured with respect to the number of isolates in each group, and the number of isolates in each group is also provided. The red dashed line denotes resistance with an azithromycin MIC > 0.25 mg/L and an erythromycin zone inhibition size of <19 mm [28]. In view of the unexpected dichotomy between azithromycin and erythromycin susceptibility, a subset of five azithromycin-resistant *S. pneumoniae* isolates were re-cultured to validate the findings. An agar-gradient diffusion method using E-test strips for azithromycin susceptibility and disc diffusion and E-test strips for erythromycin susceptibility were carried out in parallel. The repeat azithromycin MICs corresponded precisely with those previously measured for all five re-tested isolates (Appendix A) with all classed as azithromycin resistant but erythromycin sensitive as measured by the agar-gradient diffusion method using E-test strips and disc diffusion, respectively.

**Figure 6 ijms-24-16053-f006:**
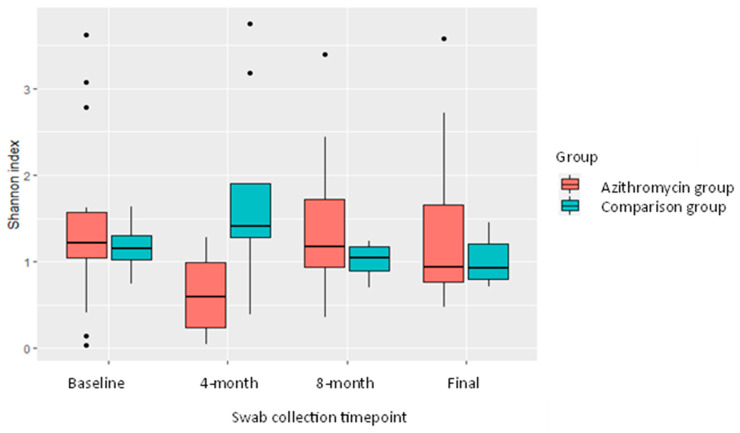
Shannon indexes of swabs from participants in both groups taken at the four swab collection timepoints. Shannon indexes have been plotted as a measure of alpha diversity for all the swabs in both the comparison and azithromycin group. The x-axis represents the time of each round of swabs. Only in the second round of swabs, when all those in the azithromycin group were actively taking azithromycin, was there a significant difference in alpha diversity, *p* = 0.02.

**Figure 7 ijms-24-16053-f007:**
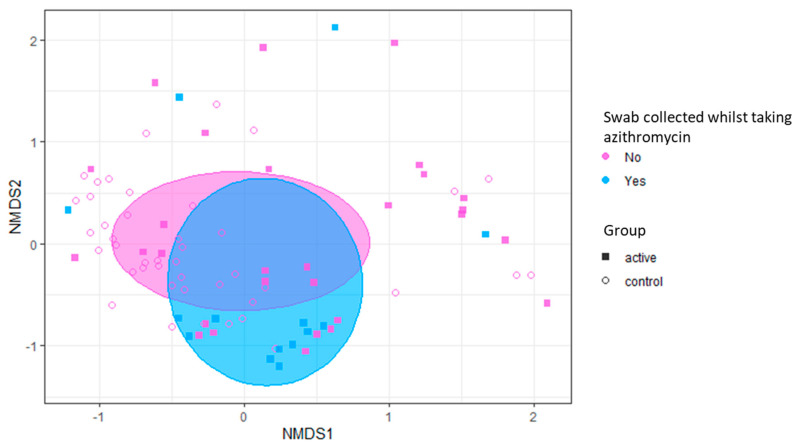
Non-metric multidimensional scaling of beta diversity for nasopharyngeal swabs collected from children taking azithromycin at the time of the swab collection. Beta diversity calculated using the Bray–Curtis dissimilarity measure and plotted using non-metric multidimensional scaling (NMDS). Coloured plots indicate samples taken from participants in the comparison group, in the azithromycin group when actively taking azithromycin or in the azithromycin group when not actively taking azithromycin. Active treatment with azithromycin had a statistically significant effect on microbiota composition when compared to all other participants. PERMANOVA; R2 = 0.026, *p* = 0.01.

**Table 1 ijms-24-16053-t001:** Baseline demographic data.

Demographic	Comparison Group N = 25 (%)	Azithromycin Group N = 25 (%)	*p* Value for *t*-Test or Fisher’s Exact Test *
Mean age in months	39.24 (SD 14.2)	50.24 (SD 24.9)	*p* = 0.063
Female	12 (48)	11 (44)	*p* = ns
Vaginal delivery	21 (84)	15 (60)	*p* = 0.113
Breast fed	13 (52)	12 (48)	*p* = ns
Microbiological diagnosis of PBB	13 (52)	20 (80)	*p* = 0.071
Premature < 37 weeks	2 (8)	4 (16)	*p* = 0.667
Ventilated for prematurity	1 (4)	2 (8)	*p* = ns
Parent smoker	6 (24)	5 (20)	*p* = 0.71
Fully immunised (UK vaccination schedule)	25 (100)	25 (100)	*p* = ns
Penicillin allergy	1 (4)	0 (0)	*p* = ns

* *t*-test conducted on continuous data and Fisher’s exact test on categorical data. ns: not significant, at an arbitrary level of 5 percent significance, two tailed.

**Table 2 ijms-24-16053-t002:** Antibiotic exposure at baseline timepoint.

Exposure	Comparison Group Median (25th–75th)	Azithromycin Group Median (25th–75th)	Statistics *
Acute courses of antibiotics before study enrolment	7 (4–10)	10 (6–13)	IRR, *p* = 0.037
Antibiotics > 2 weeks duration before study enrolment	1 (1–2)	2 (2–3)	IRR, *p* = 0.070
Time from last course of antibiotics (weeks)	11 (2–25)	5 (4–21)	*t*-test *p* = 0.953
Previous macrolide use before study enrolment	15/25 (56%)	13/25 (52%)	Fisher’s *p* = 0.776

* IRR: Incident rate ratio. *t*-test conducted on continuous data and Fisher’s exact test on categorical data.

**Table 3 ijms-24-16053-t003:** (**a**) Antibiotic susceptibility of *S. pneumoniae* isolates using disc diffusion. (**b**) Antibiotic susceptibility of *H. influenzae* isolates using disc diffusion.

(**a**)							
**Bacterial Isolate**	**Antibiotic Resistance Tested Using Disc Diffusion Unless Otherwise Stated**						
	**Azithromycin ***	**Erythromycin**		**Tetracycline**		**Oxacillin**	
*S. pneumoniae* N = 46	46	5		3		5	
(**b**)							
**Bacterial isolate**	**Antibiotic resistance tested using disc diffusion unless otherwise stated**						
	**Azithromycin ***	**Ampicillin**	**Cefuroxime**		**Chloramphenicol**		**Co-amoxiclav**
*H. influenzae* N = 54	19	12	1		2		7

* AZM—azithromycin; resistance tested using an agar-gradient diffusion method, E-test strips.

**Table 4 ijms-24-16053-t004:** Comparison of mean differences in Shannon index between the azithromycin and comparison groups.

Swab Round	Number in AZM * Group Actively Taking AZM	Mean Shannon Index AZM Group	Mean Shannon Index Comparison Group	*t*-Test *p* Value
Baseline	0	1.44	1.17	0.38
4 month	15	0.63	1.71	0.02
8 month	2	1.48	1.01	0.22
Final	4	1.33	0.97	0.22

* AZM—azithromycin.

**Table 5 ijms-24-16053-t005:** Study inclusion and exclusion criteria.

	Azithromycin Group	Comparison Group
Inclusion criteria	Age:18 months to 10 years	
	At least 1 episode of PBB over the preceding 18 months	
	Not had seasonal azithromycin prophylaxis for at least 6 months	Never been exposed to seasonal or prophylactic antibiotics
	Due to start seasonal azithromycin over the winter months—standard clinic care (local guideline [27])	Less than 3 exacerbations over the preceding 12 months. Not anticipated to require seasonal azithromycin over the study period
Exclusion criteria	Ongoing investigations for cystic fibrosis or other underlying lung disease	
	Known diagnosis of cystic fibrosis, bronchiectasis, primary immune deficiency	
	Those prescribed 3 monthly courses of intravenous antibiotics	
	Previous prescription of non-azithromycin prophylaxis	
	Bleeding disorder or anticoagulation	

## Data Availability

The data presented in this study are available on request from the corresponding author. The data are not publicly available due to privacy restrictions.

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
