# Peer review of "Seasonal Azithromycin Use in Paediatric Protracted Bacterial Bronchitis Does Not Promote Antimicrobial Resistance but Does Modulate the Nasopharyngeal Microbiome"

_ijms, 2023, doi:10.3390/ijms242216053_

Round 1

Reviewer 1 Report

Comments and Suggestions for Authors

Tha authors presented an extensive and interesting observational study oin the use of long term azytromicin in children with PBB, with a particular interest on development of antimicrobial resistance and effects on nasal microbiome.

The methods are well written and seems to be strong, and the results are complete and wide. The study limitations are well declared. I have only a suggestion on the discussion/conclusions: I think the authors can try to be more incisive, given the great amount of results obtained, on the use of seasonal azytromicin in PBB, trying to give a stronger opinion on feasibility and risks.

Author Response

Thank you very much for taking the time to review our manuscript entitled ‘Seasonal azithromycin use in paediatric protracted bacterial bronchitis does not promote antimicrobial resistance but does modulate the nasopharyngeal microbiome’. We were grateful for your helpful and supportive comments and hope to have adequately addressed the points you highlighted for amending.

Please find a detailed response below and the corresponding corrections tracked in the re-submitted files.

Point by point response to Comments and Suggestions for the authors.

Comment 1:

"I have only a suggestion on the discussion/conclusions: I think the authors can try to be more incisive, given the great amount of results obtained, on the use of seasonal azithromycin in PBB, trying to give a stronger opinion on feasibility and risks."

Response 1:

Thank you for comment with regards to the discussion/conclusion. Due to the limited power of our study imposed by the impact of the COVID-19 pandemic we are keen not to over-interpret our results, however we accept that we have perhaps been overcautious.

We have made the following changes in response to this helpful suggestions.

a) We have amended the discussion of the microbiome data as follows

(i) At baseline, Haemophilus spp. was over-abundant in the ‘to be treated’ azithromycin group, and this was significantly lower at the time of the first (3-4 month) swab (when almost all patients were still being treated in the azithromycin group), a likely beneficial effect in PBB where H. influenzae is the most frequently detected pathogen. An accompanying minor increase in Acinetobacter spp. (not shown) is likely undesirable but was likewise transient.

These changes can be found at lines 597-602

(ii) Associated with this more persistent change in beta diversity, the commensal Dolosigranulum, (related to healthy microbiota), continued at lower abundance in the treated group whereas several health associated Corynebacteria spp. were increased, suggesting a mixed impact of desirable and potentially detrimental effects

These changes are found at lines 610-614

b) In our final paragraph we have changed the text which now reads as follows:

Despite these limitations, our study informs a number of important questions regarding assessment of risk/benefit for the use of seasonal azithromycin in PBB, an intervention which is not currently supported by randomised controlled trial evidence. We found that seasonal azithromycin use was not associated with an increase in azithromycin-resistant bacterial isolates in the nasopharynx, however our finding of much higher than expected rates of background azithromycin (but not erythromycin) resistance if of concern in children with PBB (and potentially more widely) and warrants further study. This is particularly pertinent given that erythromycin is often used as a surrogate marker for macrolide resistance. The clinical impact of the perturbations in the nasopharyngeal microbiome we describe, are as yet unknown, although it is encouraging to note that many of the changes were reversible on antibiotic cessation. This has implications for the use of seasonal azithromycin more widely, as infection in many patient groups in whom azithromycin is recommended (eg COPD and adult non-CF bronchiectasis) are often concentrated in the winter period. The impact on the microbiome and resistome in other niches, such as the gut, would need to be explored before being able to fully quantify the potential risks of antimicrobial resistance secondary to seasonal azithromycin use.

This change can be found on line 647-663 the final conclusive paragraph.

Reviewer 2 Report

Comments and Suggestions for Authors

The purpose of this article was to examine the effect of the use of azithromycin for PBB in children on the development of antimicrobial resistance and on the patient’s microbiome. These are two crucial topics when dealing with broad spectrum antibiotics as cleared stated by the authors.

1.     50 children clinically diagnosed with PBB were included in the study. It is not cleared stated in the results why the comparison group (25 children) were not prescribed azithromycin as the test group. Authors state in line 103: “Azithromycin was not deemed necessary in the second group of 25 children who were used as a comparison group.” However, the reasons are not clearly stated.

2.     Authors clearly state and diagram the recruitment process for the study. They also state the previous antibiotic exposure for each group.

3.     An interesting result that should be highlighted in the abstract is the discrepancy found between laboratory methods for detecting resistant strains for azithromycin and erythromycin.

4.     Despite the limitations, which were clearly stated through the manuscript, the authors were able to report important findings that should be further studied.

Author Response

Thank you very much for taking the time to review our manuscript entitled ‘Seasonal azithromycin use in paediatric protracted bacterial bronchitis does not promote antimicrobial resistance but does modulate the nasopharyngeal microbiome’. We were grateful for your supportive comments and hope to have adequately addressed the minor points you highlighted for amending.

Please find a detailed response below and the corresponding corrections tracked in the re-submitted files.

Point by point response to Comments and Suggestions for the authors.

Comment 1:

  • 50 children clinically diagnosed with PBB were included in the study. It is not cleared stated in the results why the comparison group (25 children) were not prescribed azithromycin as the test group. Authors state in line 103: “Azithromycin was not deemed necessary in the second group of 25 children who were used as a comparison group.” However, the reasons are not clearly stated.

Response to comment 1:

  • Many thanks for addressing this point that was not clear in this section. We have included a qualification for this statement.
  • “ This was due to the fact they had not had more than three exacerbations over the preceding 12-months as is outlined in the local guideline for the management of PBB (referenced)”

This change can be found on line 105

Comment 2

  • An interesting result that should be highlighted in the abstract is the discrepancy found between laboratory methods for detecting resistant strains for azithromycin and erythromycin.

Response to comment 2:

  • We also feel this is an interesting result and have amended the abstract to include this.
  • “The discrepancy between laboratory detection of azithromycin and erythromycin resistance in the S. pneumoniae isolates requires further investigation”.

This change can be found on line 41